# Utilizing a Nordic Crosswalk for Occupational Coding in an Analysis on Occupation-Specific Prolonged Sickness Absence among 7 Million Employees in Denmark, Finland, Norway and Sweden

**DOI:** 10.3390/ijerph192315674

**Published:** 2022-11-25

**Authors:** Svetlana Solovieva, Karina Undem, Daniel Falkstedt, Gun Johansson, Petter Kristensen, Jacob Pedersen, Eira Viikari-Juntura, Taina Leinonen, Ingrid Sivesind Mehlum

**Affiliations:** 1Finnish Institute of Occupational Health, 0032 Helsinki, Finland; 2National Institute of Occupational Health (STAMI), 0363 Oslo, Norway; 3Unit of Occupational Medicine, Institute of Environmental Medicine, Karolinska Institute, 17177 Stockholm, Sweden; 4National Research Centre for the Working Environment, 2100 Copenhagen, Denmark; 5Institute of Health and Society, University of Oslo, 0450 Oslo, Norway

**Keywords:** excess fraction, mental disorders, musculoskeletal diseases, register study, work disability

## Abstract

We identified occupations with a high incidence of prolonged sickness absence (SA) in Nordic employees and explored similarities and differences between the countries. Utilizing data from national registers on 25–59-year-old wage-earners from Denmark, Finland, Norway and Sweden, we estimated the gender- and occupation-specific age-adjusted cumulative incidence of SA due to any cause, musculoskeletal diseases and mental disorders. To increase the comparability of occupations between the countries, we developed a Nordic crosswalk for occupational codes. We ranked occupational groups with the incidence of SA being statistically significantly higher than the population average of the country in question and calculated excess fractions with the employee population being the reference group. We observed considerable occupational differences in SA within and between the countries. Few occupational groups had a high incidence in all countries, particularly for mental disorders among men. In each country, manual occupations typically had a high incidence of SA due to any cause and musculoskeletal diseases, while service occupations had a high incidence due to mental disorders. Preventive measures targeted at specific occupational groups have a high potential to reduce work disability, especially due to musculoskeletal diseases. Particularly groups with excess SA in all Nordic countries could be at focus.

## 1. Introduction

Sickness absence (SA) is a major public health and economic burden to society [1]. It also has a substantial impact on employers, workplaces, employees and their families [1,2]. An increasing trend in the absence from work has been observed in Europe during 2008–2020 [3]. The Nordic countries have a higher life expectancy and fewer years lived with disability as compared to the global estimates [4]. However, compared to the other European countries, the level of prolonged SA is high in Norway and Sweden and relatively low in Denmark and Finland [5,6]. The reduction of work disability, and the resulting absence, is a priority for the Nordic countries, as well as most of the Western countries [5,7].

Sickness absence is a complex phenomenon, characterized by an interplay of several factors. In general, the determinants of SA can be categorized into three major groups: micro (e.g., individual), meso (e.g., occupation, industry, workplace) and macro (e.g., economic development, social security system, practices and legislation, composition of the labor force) level factors [1]. Previous research demonstrated that SA is unequally distributed across different population groups with a noticeable variation by age, gender, geographical region and occupational class [8,9,10,11].

Among the potential determinants of SA, work environment plays an important role. SA is typically viewed as a consequence of ill-health or injuries caused by a hazardous work environment [12]. Earlier studies reported that about 36–45% of long-term SA in the general population could be attributed to a poor work environment [13,14]. According to a Dutch study, 21.5% of the SA in the working population can be considered possibly avoidable, though only 13.6% of SA are attributable to work-related factors [15]. Globally, about 90 million disability-adjusted life years were attributable to occupational exposures in 2016 [16], justifying the need for prevention.

The existing evidence on the associations between work environment and SA is predominantly based on studies examining one or a limited set of exposures, neglecting multiple exposures, multifactorial associations and interactions between the exposures [17,18]. Occupation is a composite indicator of socioeconomic position, which reflects both educational level and income potential [19]. Moreover, occupation may be a source of multiple harmful exposures at work (e.g., physically and mentally demanding work), as well as their co-occurrence and a determinant of ill-health behavior. Furthermore, occupations may differ regarding the possibilities for people with health problems to remain at work. Studies on the associations between occupation and SA have typically explored the differences in SA between upper-level non-manual employees, lower-level non-manual employees and manual workers or examined risk factors for SA in a specific occupational group [8,9,20,21,22,23,24]. 

Despite a considerable contribution of occupational factors to SA, these factors can only partially explain the SA phenomenon. National social security systems, practices and policies have been found to have an impact on the rate and length of SA [25,26,27,28,29]. More generous policies on SA benefits make it easier for an employee to stay at home long enough when medically necessary. However, cross-country comparative studies on SA are rare [25,30,31,32] and to our knowledge, there are no comparative studies on the occupational occurrence of SA. Earlier studies were typically based on survey data and were focused on any SA (at least one day). In general, the scarcity of cross-country comparative research on SA can partly be due to insufficient availability of comprehensive, reliable and comparable data on SA. The Nordic countries are relatively similar in the composition of the labor force and have a long tradition in the establishment of administrative registers and longitudinal data sets, which provide unique opportunities to carry out register-based research on work disability. However, comparable data regarding SA in the Nordic countries based on the registers are also limited [6,33].

Understanding the variation in SA across more specific occupational groups and exploring similarities and differences in occupation-specific SA within and between countries could reveal possibilities for more targeted and focused prevention of work disability and preterm exit from paid employment. However, systematic knowledge on differences in SA across a large range of occupational groups is limited. 

In the current study, we aimed to identify occupations with a high incidence of all-cause and cause-specific prolonged SA among employees in Denmark, Finland, Norway and Sweden and to explore similarities and differences between the countries. We developed a Nordic crosswalk for occupational coding to be used in this and future cross-country comparative studies to increase the comparability of occupational groups across the countries. We hypothesize that occupational exposures play a dominant role in occupational groups with a high incidence of prolonged SA in most of the studied countries. A high incidence of prolonged SA in occupational groups observed in one country only will suggest that the SA in this case is more attributed to the workplace level, societal and/or structural factors than occupational exposures.

## 2. Material and Methods

### 2.1. Setting and Data Sources

We used individual-level data from four national register-based cohorts: the Danish cohort of employees in 2014, the Finnish Nationwide Working-Age Cohort 2013, the Norwegian Working-Age Cohort, and the Swedish Work, Illness, and Labor Market Participation Cohort. The observational year was 2015 for all four countries. When selecting the year for the analyses, we also took into consideration that the national classification codes of occupations are most comparable with the ISCO-88 (COM) classification.

The Danish cohort is drawn from the Danish register-based labor force statistics (RAS) and contains all Danes from 18 to 65 years of age, with a main occupation ultimo November 2014 (N = 2,162,390). The RAS sample is linked with the Danish Register for the Evaluation of Marginalization (DREAM). The DREAM contains individual and weekly information on all Danes, receiving any major social benefits from 1991 and onward—from which we used the 2015 data. The DREAM contains information on payments of social benefits concerning, e.g., SA, unemployment, early retirement pensions, education and parental leave, and additional information on emigration. Information on death was gained from the Danish death register. 

The Finnish cohort consists of a 70% nationally representative random sample of the Finnish working age population (18–70 years old) living in Finland on 31 December 2013 (N = 2,617,963). The data include information on compensated SA and national pensions obtained from the Finnish Social Insurance Institution (SII), on employment and earnings-related pensions from the Finnish Centre for Pensions (FCP) and on sociodemographic factors, including occupational titles obtained from the Finnish Longitudinal Employer–Employee Data (FLEED) of Statistics Finland. Information on date of death was obtained from the Population Census register and provided by the SII. 

The Norwegian cohort consists of all individuals born 1930–1992, who resided in Norway during 2000–2010, i.e., everyone who was 18–70 years during this period (N = 3,898,166). The data include information on compensated SA and national pensions and benefits obtained from the event database FD-Trygd of Statistics Norway, on SA diagnoses obtained from the sickness absence registries of the Norwegian Labor and Welfare Administration (NAV), and on employment, occupational titles, sociodemographic factors, and death obtained from Statistics Norway. 

The Swedish cohort consists of all individuals born 1941–1989 and residing in Sweden in 2005 and aged 16–64 that year (N ≈ 5.4 million). The data include information from several registers held by Statistics Sweden, such as demographic information and date of death from the Swedish register of the total population and the Longitudinal Integrated Database for Health Insurance and Labor Market Studies (LISA), information on occupational titles from the Swedish occupational register, and on SA from the register Micro-Data for Analysis of the Social Insurance System (MiDAS).

### 2.2. Study Populations

For all countries, we included individuals who were employed wage-earners according to their main economic activity and socioeconomic status and aged 25–59 at baseline (i.e., on the last day of 2014). We excluded those who were self-employed, or already had an ongoing compensated SA spell or received a full pension (full disability pension, other early retirement pensions, or old-age pension) at baseline, or had emigrated or died by the end of 2015. We also excluded persons who did not have information on the occupational title. 

The study populations for Denmark, Finland, Norway and Sweden consisted of 1,828,019 (51.1% women), 1,234,445 (52.2% women), 1,516,430 (46.7% women) and 2,375,294 (51.5% women) persons, respectively; in total, 6,954,188 persons.

### 2.3. Sickness Absence Benefits

Denmark: SA benefit is possible only for individuals with a legal residence in Denmark, and they are required to have had at least 74 h of labor, within eight weeks of continued employment at the respective employer. An employer can apply for SA benefit for a sick-listed employee after 30 days of SA (full-time or part-time). The SA benefit is typically paid to the employer by the municipality as a compensation for the continuous salary payment to a sick-listed employee. There is no limit to the duration of the SA benefits, but payments are typically stopped after 22 weeks of SA, when a plan of labor return must be ready. In Denmark there is no diagnosis registered with the SA period, only the first and last day of SA.

Finland: Eligibility for a compensated sickness benefit requires that the claimant resides permanently in Finland. After a waiting period of 10 weekdays, during which the employer typically pays salary to the sick-listed person, SA is compensated by the SII as part of the national social security system until a maximum of 300 weekdays (Sundays excluded). The register of the SII provides information on the start and end dates as well as primary diagnoses for all full-time SA spells exceeding the waiting period and for all part-time SA spells. The SA diagnoses, set by the treating physician, are classified according to the International Statistical Classification of Diseases and Related Health Problems, Tenth Revision (ICD-10, Finnish version of ICD-classification 1996). 

Norway: Individuals who have been in paid employment for at least four weeks prior to the SA are entitled to SA benefits. SA benefits are covered by the employer for the first 16 calendar days of SA. After this, the employer is compensated by NAV until a maximum of 52 weeks of SA, and the episodes are registered in the sickness benefits database, which includes complete information on all compensated SA spells. The register provides information on start and end dates for full-time and part-time SA periods, as well as primary diagnoses, based on the doctor’s notification, coded according to the International Classification of Primary Care, second edition (ICPC-2). The main diagnostic groups in ICPC-2 are comparable to those in ICD-10. 

Sweden: To be entitled to SA benefit, a person has to be employed, or if unemployed, has had a job prior to unemployment. The first day is a qualifying day for which no salary or benefit is paid to the employee. The employer pays compensation from day two until day 14. From day 15, the sickness benefit is paid by the Swedish Social Insurance Agency, and therefore recorded. In general, there is no limit to the duration of the SA benefits, but after receiving the sickness benefit for 364 days, the employee needs to reapply to continue receiving it. Information in the register includes dates for the start and end of the full-time and part-time SA periods as well as SA diagnoses according to ICD-10.

A recent review on national SA policies in European countries provides a more detailed comparison of SA systems of Denmark, Finland, Norway and Sweden [29].

### 2.4. Outcome

The outcome of this study was the first SA episode (either full- or part-time) of any cause that started during 2015 and lasted for at least 30 days (one-year cumulative incidence of prolonged SA, later referred to as “incidence of SA”). In addition to all-cause SA, we examined the two largest disease groups separately, namely diseases of the musculoskeletal system and connective tissue (ICD-10 M00–M99) and mental and behavioral disorders (F00–F99). Information on the SA diagnosis was available for Finland, Norway and Sweden. 

### 2.5. Occupation

Occupations were classified according to national variants of the European version of the International Standard Classification of Occupations (ISCO (COM)): Statistics Denmark’s Classification of Occupations (DISCO-08) [34], the Classification of Occupations 2001 by Statistics Finland (FISCO-01) [35], the Norwegian Standard Classification of Occupations 1998 by Statistics Norway (STYRK-98) [36] and the Swedish Standard Classification of Occupations 1996 by Statistics Sweden (SSYK-96) [37]. All national classifications were based on the European version of the International Standard Classification of Occupations (ISCO (COM)). DISCO-08 is a revised version of DISCO-88/ISCO-88 (COM). In the current study, to improve comparability between the occupational codes of all four countries, DISCO-08 was converted into DISCO-88 using a crosswalk provided by Statistics Denmark and further manually checked by a researcher (JP). For some occupational codes (approx. 18%), the DISCO-08-DISCO-88 crosswalk does not have a one-to-one conversion solution [38] and they were coded manually.

### 2.6. Development of the Nordic Crosswalk

In order to make the occupational titles from the four countries comparable, we created a Nordic crosswalk, with common Nordic occupational codes, on the basis of ISCO-88 (COM) (Appendix A). ISCO-88 (COM) includes 376 4-digit occupational codes (including “army personnel” group). 

DISCO-88 contains 372 4-digit occupational codes and is better comparable to ISCO-88 (COM) than the occupational classifications of the three other Nordic countries. FISCO-01, SSYK-96 and STYRK-98 include 351, 354 and 350 4-digit occupational codes, respectively. There were, in total, 484 national occupational codes. Some occupational codes in the national classifications were not found in ISCO-88 (COM). Similarly, there were occupational codes in ISCO-88 (COM), which were not found in the national classifications (Appendix A).

First, each national occupational classification was compared with ISCO-88 (COM) to identify occupational groups, which (1) could be exactly matched based on title, (2) not exactly matched based on title and (3) unique for either national classification or ISCO-88 (COM). Second, we combined the crosswalks between the classification of each country and ISCO-88 (COM) into one crosswalk. Third, for unmatched occupational codes, we critically reviewed the descriptions of the occupations under the codes and suggested a common Nordic occupational code. 

For several occupational codes belonging to “Professionals”, “Associate professionals” or “Service workers”, it was difficult to find a corresponding match between the national occupational codes. For these occupational groups it was allowed that professionals and associate professionals, as well as associate professionals and service workers, would receive the same Nordic code, if the tasks performed, exposure profiles and work environment were considered comparable. Such a rule was applied for the “primary education teaching professionals and associate professionals” and “pre-primary education teaching professionals and associate professionals” (Nordic codes 2330 and 3320, respectively). In the Norwegian and Swedish classifications “police officers” had a higher hierarchical code and were combined with “police inspectors and detectives”, while in the Danish and Finnish classifications, these two occupational groups received different hierarchically distinguished codes similar to those used in the ISCO-88 (COM) classification. In the crosswalk, “police inspectors and detectives” and “police officers” received Nordic codes 3450 and 5162, respectively. For Norway and Sweden, Nordic codes 5162 included both “police inspectors and detectives” and “police officers”.

Two ISCO-88 (COM) occupational codes (3330 “Special education teaching associate professionals” and 5142 “Companions and valets”) were found only in the DISCO-88 classification, but not in the classifications of the other countries. In addition, for eight Nordic codes, the corresponding occupational code in the national classification of one or two of the Nordic countries was not found.

### 2.7. Occupational Groups Used in the Analyses

We examined similarities and differences in the distributions of the common Nordic occupational codes in the study population across the four countries. We observed relatively large differences for some of the occupational codes. We also found that the size of some occupational groups was small. To further increase the comparability of occupational groups across the countries, we merged 124 common Nordic codes with large differences in the size across the countries into 44 aggregated occupational groups (Appendix A). 

For further analyses, we included only occupational groups with 100 or more workers per group. Into the comparative analysis of all-cause SA between four countries, we included 170 occupational groups in men and 136 in women. Into the comparative analysis of cause-specific SA between three countries, we included 174 and 146 occupational groups in men and women, respectively.

To summarize the results, the following major occupational groups based on the first digit of the Nordic codes were used: 1- Legislators, senior officials and managers; 2- Professionals; 3- Associate professionals; 4- Clerks; 5- Service and care workers, and shop and market sales workers; 6- Skilled agricultural and fishery workers; 7- Craft and related trades workers; 8- Plant and machine operators and assemblers; 9- Elementary occupations.

### 2.8. Statistical Analyses

We used a general linear model to estimate the age-adjusted one-year cumulative incidence of prolonged all-cause and cause-specific SA as well as their 95% confidence intervals (95% CI) for each occupational group. We ranked occupational groups based on SA, with the lowest rank being assigned to the group with the highest SA incidence. Occupational groups with the same SA incidence received the same rank. To control for the large variation in the size of the occupational groups, only those with SA incidence statistically significantly higher than in the employee population of the country in question were ranked. The ranking was carried out separately for each country, gender and SA outcome.

For the ranked occupational groups, excess fractions (EFs) [39,40] and their 95% CIs were calculated for each SA outcome. The EF was calculated using the following formula: EF = (I_o_ − I_g_)/I_o_, (1)
where I_o_—is the incidence of SA in a specific occupational group and I_g_—is the incidence of SA in the general employee population. The values were expressed in percentages. 

The general employee population incidence was selected as the reference instead of the occupational group with the lowest SA incidence, in order to have a similar reference group for both genders and all countries, as well as to control for large differences in the overall level of SA across the countries. In this study, with EF we mean the proportion of prolonged SA that would not have occurred, had the incidence in each occupational group been as in the general employee population. We also calculated the weighted mean of the country specific EFs, using the following formular:weighted mean EF = (∑n_i_ EF_i_)/∑ n_i_,(2)
where n_i_—is the country-specific number of employees in the occupational group, EF_i_—is the country-specific EF value for this occupational group and i corresponds to the countries contributing to the estimates. For all-cause and cause-specific SA, the weighted mean EF was calculated based on EFs of four and three countries, respectively. 

## 3. Results

Among both men and women, the age-adjusted one-year cumulative incidence of prolonged SA due to all causes and musculoskeletal diseases was highest in Norway and second highest in Sweden (Figure 1, panel A and B). The difference in all-cause SA between Finland and Denmark was small. The age-adjusted incidence of SA due to mental disorders was highest in Sweden and second highest in Norway (Figure 1, panel B). In all countries, women had a higher incidence of SA than men, with the gender difference being particularly large in Norway and Sweden. 

### 3.1. Nordic Crosswalk

Based on occupational codes of all four countries, we created a total of 278 common Nordic occupational codes (excluding army personnel group). The Nordic crosswalk is shown in the Appendix A. 

### 3.2. Occupational Differences in All-Cause SA within Denmark, Finland, Norway and Sweden

In both genders, we found more occupational groups with SA incidence being higher than the population average in Norway than in the three other countries (Appendix A, dark-colored squares). Table 1 and Table 2 show the occupational groups with rank 1–10, based on prolonged all-cause SA, in at least one of the four countries for men and women, respectively. Overall, 29 occupational groups in men and 27 in women were among the top 10 ranked groups in at least one of the countries. Among men, the highest EFs within the countries were found for “insulation workers” (52.9%, Denmark), “ships’ engineers” (62.6%, Finland), “ships’ deck crews and related workers” (53.2%, Norway) and “butchers, fishmongers and related food preparers” (57.3%, Sweden). Three occupational groups—“ships’ deck crews and related workers”, ”butchers, fishmongers and related food preparers” and “roofers”—were among the top 10 ranked occupations in three countries (Table 1). 

The occupational groups with the highest EFs within the countries among women were “car, taxi, motorcycle and van drivers” (54.0%, Denmark), “veterinary assistants” (60.0%, Finland), “heavy truck and lorry drivers” (43.4%, Norway), and “vehicle, window and related cleaners” (49.5%, Sweden) (Table 2). In all countries, except Sweden, the occupational groups with the highest EF were relatively small (<500 workers). Three occupational groups—“bus and tram drivers”, “transport conductors” and “nursing and care assistants”—were among the top 10 ranked occupations in all four countries. In addition, “heavy truck and lorry drivers” and ”painters, varnishers and related workers” were among the top 10 ranked groups in three countries (Table 2). 

### 3.3. Occupational Differences in SA Due to Musculoskeletal Diseases and Mental Disorders within the Countries

In both genders, the largest number of occupational groups with the incidence of SA due to musculoskeletal diseases being higher than the population average was observed in Norway (Appendix A). The largest number of occupational groups with the incidence of SA due to mental disorders being higher than the population average was observed in Sweden (Appendix A). 

In all three countries with available diagnosis-specific information, two occupational groups among men—”butchers, fishmongers and related food preparers” and “glaziers” —received a rank between 1 and 10 based on SA due to musculoskeletal diseases (Table 3). Three occupational groups—“telephone switchboard operators”, “professionals and associate professionals in social work” and “nursing and care assistants”—were among the top 10 ranked groups based on SA due to mental disorders (Table 4).

Among women only the group of “wood-products machine and plants operators” received a rank between 1 and 10 based on SA due to musculoskeletal diseases in all three countries (Table 5). Among the top 10 ranked groups based on SA due to mental disorders, three occupational groups—“transport conductors”, “religious professionals” and “professionals and associate professionals in social work”—were observed in all three countries (Table 6).

### 3.4. Occupational Differences in All-Cause and Cause-Specific SA across the Nordic Countries

A total of 29 occupational groups among men and 14 among women had a high EF of prolonged all-cause SA in all four countries and were thus included in a further comparison of these groups across the Nordic countries (Table 7 and Table 8). The weighted mean EF value varied between 16.9% (“electricians”) and 44.0% (”butchers, fishmongers and related food preparers”) in men and between 18.2% (“professionals and associate professionals in social work”) and 48.0% (”bus and tram drivers”) in women. Among both men and women, most of the occupational groups within the major occupational groups of “craft and related trades workers”, “plant and machine operators and assemblers”, “elementary occupations” and “service and care workers, and shop and market sales workers” had EF values for all-cause SA being 20% or higher (Table 7 and Table 8).

In both genders, a larger number of occupational groups with a high EF of SA in all three Nordic countries was observed for musculoskeletal diseases than mental disorders (27 vs. 5 in men and 18 vs. 10 in women). Among men and women, the highest weighted mean EF of SA due to musculoskeletal diseases was observed for “butchers, fishmongers and related food preparers”, being 68.2% (66.1–69.8) and 60.3% (48.7–66.4), respectively. ”Nursing and care assistants” (57.7%, 56.1–59.5) and ”religious professionals” (54.9%, 50.0–58.3) had the highest weighted mean EF of SA due to mental disorders in men and women, respectively. 

## 4. Discussion

In this study, utilizing data from national registers, we looked at occupations with a high incidence of all-cause and cause-specific prolonged SA in four Nordic countries and explored similarities and differences between the countries. To increase the comparability of occupational groups between the countries, we developed the Nordic crosswalk for occupational codes. We observed considerable occupational differences in prolonged SA within and between the Nordic countries. In both men and women, Norway had the largest number of occupational groups with a high SA incidence of any cause and due to musculoskeletal diseases, while Sweden had the largest number of occupations with a high SA incidence due to mental disorders. A relatively small number of occupational groups was observed with a high incidence of SA in all studied countries, particularly for mental disorders among men. In each country, manual occupational groups were in dominance among groups with an incidence of SA of any cause and due to musculoskeletal diseases being higher than the population average, while occupational groups with a high SA incidence due to mental disorders were typically service occupations.

Few previous studies have compared the patterns of the distribution of SA across different countries [25,30,31,32]. To our knowledge, differences in cause-specific SA overall or occupational differences in SA between the Nordic countries have not been reported. We identified common elements in the register data on SA of the four Nordic countries and selected prolonged SA (at least 30 compensated SA days) as a harmonized and standardized outcome to enable a cross-country comparison. We found a large variation in the overall SA incidence between the Nordic countries, particularly among women. Our overall country-specific results regarding all-cause SA are in line with those previously reported [6]. Differences in SA between the countries could be, to some extent, attributed to differences in the social security system, labor market culture and the composition of the workforce. A study by Osterkamp and Röhn [41] on SA of 20 countries found that the generosity of social security systems and the strictness of employment protection are the most important explanatory factors of international variation in SA rates. Although Nordic countries have been considered similar, they differ with regard to the generosity of sickness absence benefits, occupational health services, management of long-term SAs, expenditures of retirement and right to “fire” employees, as well as macroeconomic conditions [29]. 

The previous studies on occupational differences in prolonged SA usually used rather general occupational classes or focused on the determinants of SA for specific occupations [8,9,20,21,22,23,24]. In contrast, we used a large set of more specific occupational groups to examine similarities and differences in SA within and between the countries. Due to large differences in the levels of SA across the countries, the comparison in this study was made in relative terms. We used EF to identify occupational groups with excess SA as compared to the general employee population and to explore occupational differences in prolonged SA. In all studied countries, a high excess SA of any cause as well as due to musculoskeletal diseases was mostly observed in occupational groups belonging to the following larger main categories “craft and related trades workers”, “plant and machine operators and assemblers”, “elementary occupations” and “service and care workers, and shop and market sales workers”. These data are in line with the findings of previous studies, especially regarding occupations with greater physical demands [10,21,23,42]. 

The high excess SA due to mental disorders was most observed in occupational groups belonging to the following larger main categories “service and care workers, and shop and market sales workers”, “clerks” and “associate professionals”. These groups have usually higher qualification levels than manual occupations and are characterized by high psychological demands [10,43]. 

Even though there were differences in occupations with a high incidence of SA between the Nordic countries, several occupational groups appeared among the ranked occupations based on all-cause and cause-specific SA in all studied countries. For example, among men in all four countries, “mail carriers and sorting clerks”, “nursing and care assistants”, construction workers (e.g., “builders, bricklayers and stonemasons”, “insulation workers” and “plumbers and pipe fitters”), professional drivers (“car, taxi, motorcycle and van drivers”, “bus and tram drivers” and “heavy truck and lorry drivers”) and laborers in mining, construction, manufacturing and transport had an EF of all-cause SA above 30%. Among women, “mail carriers and sorting clerks”, “transport conductors”, “nursing and care assistants”, “home-based personal care workers”, “wood-products machine and plants operators” and “bus and tram drivers” had an EF of all-cause SA above 30% in all four countries.

The EFs, calculated based on register data in our study, can also be observed as an alternative to the etiologic fraction [39,40,44] and thereby provide a conservative estimate of the prevention potential [44]. In the current study, the EF could be interpreted as the proportion of SA attributable to a specific occupational group. However, the observed occupational differences in excessive SA could also be due to selection into the occupation. Our findings suggest that among both men and women, at least 20% of all-cause SA in the Nordic countries could be attributed to occupations belonging to the major occupational groups “craft and related trades workers”, “plant and machine operators and assemblers”, “elementary occupations” and “service and care workers, and shop and market sales workers”. However, the weighted mean EF of all-cause SA values tended to be higher among men than women for a similar occupation. In general, the weighted mean EF values were higher for cause-specific than all-cause SA, suggesting that the cause-specific SA could be attributed to an occupation to a larger extent than the all-cause SA. Furthermore, occupations with excess SA were often observed in only one or some of the countries. Therefore, occupational groups with consistent excess SA across the countries will potentially benefit the most from interventions aiming at reduced physical and/or psychosocial workload.

### Strengths and Limitations of the Study

Our study has several strengths. There was a good representativeness of the employee population of Denmark, Finland, Norway and Sweden since we utilized large nationally representative register-based samples from these countries. International collaborative studies can produce more generalizable information than country-specific studies [30]. To enable cross-country comparisons, we used the developed Nordic occupational crosswalk, as well as harmonized and standardized SA outcomes. Furthermore, we limited the comparison to wage earners, as the eligibility and use of SA vary for both self-employed workers and unemployed. Moreover, we explored occupational differences in age-adjusted SA incidence to control for age differences across the occupations. Finally, we used the EF of SA as a measure for cross-country comparison to control for differences in overall SA incidence between the countries. 

One of the largest study limitations is the lack of cause-specific SA data for Denmark. The episodes of SA and diagnosis are not recorded simultaneously in Denmark and due to this, the possibility of direct merging of such information does not exist. Furthermore, during the development of the Nordic crosswalk, we faced challenges in the one-to-one matching of national occupational codes, since all of them were not found in ISCO-88 (COM). To increase the comparability across the countries, we combined the occupations with unmatched codes with occupations with similar descriptions and assigned to them a common Nordic occupational code. When the developed crosswalk was applied to the current study population, we observed large differences in the occupational distributions across the countries, with several occupational groups, especially among women, being of a small size. In the Danish study population, occupations were coded using DISCO-08 and converted into DISCO-88. Several occupational codes were lost because the DISCO-08–DISCO-88 crosswalk does not have a one-to-one conversion solution. To minimize observed differences in the composition of the occupational groups, we created 44 aggregated groups, which were used in the analyses. The SA incidence and thus EFs might be overestimated for the relatively small occupational groups. To minimize biased results, the EFs were calculated only for occupations with SA incidence being statistically significantly higher than the population average of the country in question.

The observed large variations in the occupational groups with excessive SA incidence in different countries could partly be due to cross-country structural and compositional differences, which we were not able to control for. We made analyses for men and women separately and controlled for age differences in the composition of occupations within the countries by using the age-adjusted incidence of SA. However, we were not able to control for age differences between the countries within the same occupation. Furthermore, we were not able to control for differences in social security and work disability practices between the public and private sectors. The occupational coding systems are not able to differentiate occupations of the public and private sectors. 

The data used for the analyses are relatively old; however, the advantage is that we have harmonized data across the countries. Even though the incidence of prolonged SA may change over time, the occupational differences in the incidence are known to be stable. In the current study, we explored occupational difference using a relative measure (excess fraction) instead of absolute (incidence). Our main finding regarding occupations with an excessive incidence of prolonged SA is likely to apply more widely than to a single point of time and thus is relevant also for the present time.

## 5. Conclusions

We identified several occupational groups with a high SA incidence that have also been reported earlier in the literature. The number of occupational groups with an excessive incidence of SA due to musculoskeletal diseases in all three countries was larger than that due to mental disorders. Our main findings regarding occupational groups with an excessive incidence of prolonged SA suggest that occupational exposures contribute more to SA due to musculoskeletal diseases than due to mental disorders. The SA due to mental disorders is likely to be more attributable to the workplace level, societal and/or structural factors than occupational exposures. Our results indicate that occupational preventive measures have a high potential for the prevention of cause-specific work disability, especially due to musculoskeletal diseases. The prevention of work disability could be targeted at the identified occupational groups with excess prolonged SA in all Nordic countries. Future studies need to explore to what extent occupational differences in SA in these occupational groups are attributed to work-related factors. The developed Nordic crosswalk can be useful in future cross-country comparative studies on occupational health, despite its limitations. 

## Figures and Tables

**Figure 1 ijerph-19-15674-f001:**
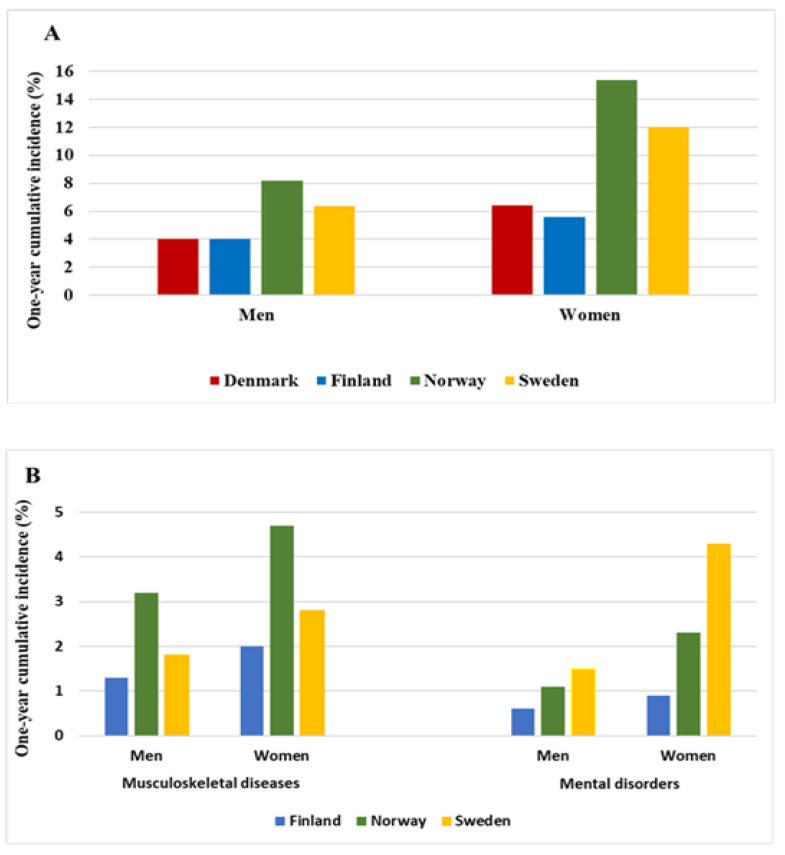
Age-adjusted one-year cumulative incidence of sickness absence among men and women by country, (**A**): Age-adjusted one-year cumulative incidence of all-cause sickness absence; (**B**): Age-adjusted one-year cumulative incidence of cause-specific sickness absence.

**Table 1 ijerph-19-15674-t001:** Occupational groups with rank 1–10 based on the excess fraction (EF, %) of the age-adjusted one-year cumulative incidence of prolonged all-cause sickness absence among men in four Nordic countries.

	Occupational Group	Denmark		Finland		Norway		Sweden	
		EF	Rank	EF	Rank	EF	Rank	EF	Rank
7134	Insulation workers	52.9	1			42.0	6		
3141	(FI; SE) Ships’ engineers			62.4	1				
8340	Ships’ deck crews and related workers	44.4	5	52.2	6	53.2	1		
7411	Butchers, fishmongers and related food preparers			57.7	3	41.4	8	57.3	1
7131	Roofers	48.1	2			49.8	2	48.8	4
5112	(FI) Transport conductors			62.0	2	41.1	9		
0021	(FI; SE) Travel attendants and guides							56.8	2
8271	(FI) Meat- and fish-processing-machine operators	46.7	3					44.3	7
8323	Bus and tram drivers					49.6	3	46.5	6
7121	Builders, bricklayers and stonemasons	46.7	3						
9140	(FI) Vehicle, window and related cleaners	39.4	10					55.5	3
5163	Prison guards	46.0	4	55.7	4				
7216	(DK; FI; NO; SE) Underwater workers					48.6	4		
8333	Crane, hoist and related plant operators					45.9	5		
5130	Nursing and care assistants	44.4	5	43.4	9				
3143	Aircraft pilots and related associate professionals			53.2	5				
7132	(NO) Floor layers and tile setters							46.8	5
7140	Painters, varnishers and related workers	43.7	6						
8320	Car, taxi, motorcycle and van drivers	42.9	7			41.9	7		
0026	Metal- and mineral-products machine operators			47.8	7				
7211	(NO) Metal moulders and coremakers					41.9	7		
9160	(FI) Garbage collectors and related labourers					41.0	10	43.9	8
7136	Plumbers and pipe fitters	41.2	8						
8312	(DK; NO) Railway brakers, signallers and shunters	40.3	9	47.3	8				
9310	Mining and construction labourers	40.3	9						
7135	(FI) Glaziers	40.3	9					43.7	9
9210	(FI; NO) Agricultural, fishery, forestry and related labourers	40.3	9						
0028	Rubber- and plastic-products machine operators			43.3	10				
7120	(FI) Concrete placers, plasterers, concrete finishers and related workers							42.9	10

Capital letters in parentheses before the name of an occupational group correspond to the abbreviated country name and indicate a small size of the occupational group (<500 workers) in the country.

**Table 2 ijerph-19-15674-t002:** Occupational groups with rank 1–10 based on the excess fraction (EF, %) of the age-adjusted one-year cumulative incidence of prolonged all-cause sickness absence among women in four Nordic countries.

	Occupational Group	Denmark		Finland		Norway		Sweden	
		EF	Rank	EF	Rank	EF	Rank	EF	Rank
8320	(DK) Car, taxi, motorcycle and van drivers	54.0	1						
3227	(FI) Veterinary assistants			60.0	1				
8324	(DK; NO) Heavy truck and lorry drivers	45.3	4			43.5	1	36.7	6
9140	Vehicle, window and related cleaners							49.5	1
8323	Bus and tram drivers	50.8	2	50.3	4	42.2	3	47.4	2
5112	(FI; DK; NO) Transport conductors	41.3	8	55.0	2	43.4	2	43.0	3
4142	Mail carriers and sorting clerks			51.3	3				
5163	(FI) Prison guards	46.7	3						
9160	Garbage collectors and related labourers					37.1	4		
3132	Broadcasting and telecommunications equipment operators							40.4	4
7140	Painters, varnishers and related workers	43.9	5			33.9	6	34.3	7
7411	Butchers, fishmongers and related food preparers			48.3	5				
8130	Glass, ceramics, paper and related plant operators					35.1	5		
0021	Travel attendants and guides							37.4	5
5130	Nursing and care assistants	42.8	6	43.2	10	33.3	8	33.6	8
6130	Crop and animal producers			48.1	6				
8120	(FI; NO) Metal-processing plant operators			46.5	7				
5162	Police officers			46.2	8				
7212	Welders and flame cutters					33.5	7		
8150	Chemical-processing-plant operators	42.3	7						
9310	Mining and construction labourers					33.2	9		
7136	Plumbers and pipe fitters			45.1	9				
5133	Home-based personal care workers							30.9	9
8240	(NO) Wood-products machine and plants operators	39.1	9						
7420	(DK: NO; SE) Wood workers and related machine operators	37.9	10						
0027	Chemical-products machine operators					32.6	10		
8271	Meat- and fish-processing-machine operators							30.8	10

Capital letters in parentheses before the name of an occupational group correspond to the abbreviated country name and indicate a small size of the occupational group (<500 workers) in the country.

**Table 3 ijerph-19-15674-t003:** Occupational groups with rank 1–10 based on the excess fraction (EF, %) of the age-adjusted one-year cumulative incidence of prolonged sickness absence due to musculoskeletal diseases among men in Finland, Norway and Sweden.

	Occupational Group	Finland		Norway		Sweden	
		EF	Rank	EF	Rank	EF	Rank
5112	(FI) Transport conductors	70.4	1				
7131	Roofers			65.3	1	69.2	3
7411	Butchers, fishmongers and related food preparers	68.4	3	62.2	3	73.2	1
5163	Prison guards	69.4	2				
8340	Ships’ deck crews and related workers	57.4	9	63.2	2		
7132	(NO) Floor layers and tile setters					71.0	2
0026	Metal- and mineral-products machine operators	64.8	4				
7135	(FI) Glaziers	56.8	10	58.7	4	61.7	6
7121	Builders, bricklayers and stonemasons			53.9	7	62.8	4
8113	(FI) Mining and mineral-processing-plant operators	63.2	5				
8323	Bus and tram drivers			55.0	5		
9140	(FI) Vehicle, window and related cleaners					62.4	5
4142	Mail carriers and sorting clerks	61.1	6				
8320	Car, taxi, motorcycle and van drivers			54.1	6		
8311	Locomotive engine drivers	59.7	7				
7120	(FI) Concrete placers, plasterers, concrete finishers and related workers			53.3	8	60.0	7
0028	Rubber- and plastic-products machine operators	57.7	8				
8271	(FI) Meat- and fish-processing-machine operators					58.4	8
8333	Crane, hoist and related plant operators			52.5	9		
7140	Painters, varnishers and related workers					57.6	9
9130	Domestic helpers, cleaners and related workers			52.2	10		
9160	(FI) Garbage collectors and related labourers					55.3	10
7129	Carpenters and building frame and related trades workers					55.3	10

Capital letters in parentheses before the name of an occupational group correspond to the abbreviated country name and indicate a small size of the occupational group (<500 workers) in the country.

**Table 4 ijerph-19-15674-t004:** Occupational groups with rank 1–10 based on the excess fraction (EF, %) of the age-adjusted one-year cumulative incidence of prolonged sickness absence due to mental disorders among men in Finland, Norway and Sweden.

	Occupational Group	Finland		Norway		Sweden	
		EF	Rank	EF	Rank	EF	Rank
4223	(NO) Telephone switchboard operators	70.1	1	70.1	1	52.4	6
0021	(FI) Travel attendants and guides					60.3	1
3421	(FI; NO; SE) Trade brokers	66.6	2				
0035	Professionals and associate professionals in social work	49.8	9	68.2	2	55.7	4
9140	(FI) Vehicle, window and related cleaners					57.5	2
2432	(FI; NO) Librarian and related information professionals	66.1	3				
0011	Nursing and midwifery professionals and associate professionals			59.2	3		
5139	Unspecified personal care and related workers			37.2	10	56.9	3
5133	(NO) Home-based personal care workers	65.3	4	58.2	4		
5130	Nursing and care assistants	65.0	5	56.8	5	53.5	5
2460	Religious professionals	59.3	6				
5131	Child-care workers			56.7	6		
3480	Religious associate professionals	57.5	7				
3320	(FI) Pre-primary education professionals and associated professionals			45.6	7		
2445	(FI) Journalists and other writers; radio and other announcers					52.2	7
9110	Street vendors and related workers	51.8	8	39.3	8		
5163	Prison guards					47.3	8
8323	Bus and tram drivers			37.9	9		
0013	Personnel and carrier professionals					41.1	9
0019	Sculptors, painters and commercial designers	47.9	10				
4222	Receptionists and information clerks					40.5	10

Capital letters in parentheses before the name of an occupational group correspond to the abbreviated country name and indicate a small size of the occupational group (<500 workers) in the country.

**Table 5 ijerph-19-15674-t005:** Occupational groups with rank 1–10 based on the excess fraction (EF, %) of the age-adjusted one-year cumulative incidence of prolonged sickness absence due to musculoskeletal diseases among women in Finland, Norway and Sweden.

	Occupational Group	Finland		Norway		Sweden	
		EF	Rank	EF	Rank	EF	Rank
8120	(FI; NO) Metal-processing plant operators	71.9	1	54.1	5		
7212	Welders and flame cutters			60.9	1		
9140	Vehicle, window and related cleaners					65.7	1
5112	(FI; DK; NO) Transport conductors	68.5	2	51.7	8		
0028	Rubber- and plastic-products machine operators			57.8	2		
7411	Butchers. fishmongers and related food preparers	64.7	4			61.0	2
4142	Mail carriers and sorting clerks	67.0	3			53.6	7
8240	Wood-products machine and plants operators	61.0	5	56.9	3	54.2	6
7140	Painters, varnishers and related workers			51.9	6	58.1	3
8324	(DK; NO) Heavy truck and lorry drivers			56.0	4		
8323	Bus and tram drivers	53.9	9			57.9	4
6140	Motorised farm and forestry related workers					55.0	5
5162	Police officers	55.3	6				
6130	Crop and animal producers	55.2	7				
8271	Meat- and fish-processing-machine operators			51.9	7	52.5	10
9310	Mining and construction labourers	55.0	8				
5161	Fire-fighters					53.0	8
0027	Chemical-products machine operators						
9320	Manufacturing and transport labourers			51.0	9	52.5	9
3450	Police inspectors and detectives	53.6	10				
0044	Grocery and beverage machine operators			50.2	10		

Capital letters in parentheses before the name of an occupational group correspond to the abbreviated country name and indicate a small size of the occupational group (<500 workers) in the country.

**Table 6 ijerph-19-15674-t006:** Occupational groups with rank 1–10 based on the excess fraction (EF, %) of the age-adjusted one-year cumulative incidence of prolonged sickness absence due to mental disorders among women in Finland, Norway and Sweden.

	Occupational Group	Finland		Norway		Sweden	
		EF	Rank	EF	Rank	EF	Rank
3421	Trade brokers	68.5	1				
1221	Production and operations managers in agriculture, hunting, forestry and fishing			57.4	1		
3132	Broadcasting and telecommunications equipment operators					57.0	1
8150	Chemical-processing-plant operators	66.4	2				
5112	(FI; NO) Transport conductors	57.3	5	51.7	2	44.1	5
2460	Religious professionals	62.8	3	46.1	4	53.9	2
8323	Bus and tram drivers			49.3	3	46.4	4
9140	Vehicle, window and related cleaners					46.8	3
3480	Religious associate professionals	59.5	4				
0035	Professionals and associate professionals in social work	56.3	6	45.4	5	43.1	6
2431	Archivists and curators			44.5	6		
9110	Street vendors and related workers	55.5	7	39.8	7		
7129	Carpenters and building frame and related trades workers					41.0	7
5133	Home-based personal care workers	51.7	8				
8320	Car. taxi. motorcycle and van drivers			39.2	8		
5139	Unspecified personal care and related workers					38.1	8
7130	Electricians	50.9	9				
3320	Pre-primary education teaching professionals and associate professionals			33.1	9		
5163	(FI) Prison guards					37.6	9
2223	Veterinarians	49.5	10				
5169	Unspecified protective cares workers			32.3	10		
2445	Journalists and other writers; radio and other announcers					35.3	10

Capital letters in parentheses before the name of an occupational group correspond to the abbreviated country name and indicate a small size of the occupational group (<500 workers) in the country.

**Table 7 ijerph-19-15674-t007:** Weighted means of country-specific excess fractions (EF, %) of the age-adjusted one-year cumulative incidence of prolonged sickness absence (SA) among men. Only occupational groups with statistically significant excess fractions in all countries are shown.

		EF of All-Cause SA		EF of SA Due to Musculoskeletal Diseases ^1^		EF of SA Due to Mental Disorders ^1^	
Code	Title	Mean (%)	95% CI	Mean (%)	95% CI	Mean (%)	95% CI
0026	Metal- and mineral-products machine operators	25.1	23.8–26.2	40.2	38.0–42.4		
0027	Chemical-products machine operators	32.0	27.3–35.4	43.1	38.2–46.3		
0028	Rubber- and plastic-products machine operators	31.5	28.2–34.4	48.9	46.6–50.7		
0029	(NO) Book- and paper- products machine operators	30.6	22.8–36.0				
0030	Agricultural or industrial machinery fitters, and mechanical and electrical equipment assemblers	24.8	23.5–26.1	41.5	39.5–44.7		
0035	Professionals and associate professionals in social work	26.2	23.6–28.6			56.2	53.3–58.8
0044	Grocery and beverage machine operators	28.8	25.0–32.0	41.7	37.8–44.7		
4142	Mail carriers and sorting clerks	34.2	32.9–35.5	50.4	49.6–51.1		
4223	(NO) Telephone switchboard operators					57.2	49.7–61.3
5112	(FI) Transport conductors						
5122	Cooks	22.7	19.3–25.2				
5130	Nursing and care assistants	42.1	41.5–42.7	38.3	36.6–39.7	57.7	56.1–59.5
5133	(NO) Home-based personal care workers	30.8	29.7–31.9			42.6	40.0–45.0
5161	Fire-fighters			44.7	41.0–47.3		
5163	Prison guards	37.1	32.8–40.2				
5169	Unspecified protective service workers			26.4	20.8–30.4		
7121	Builders, bricklayers and stonemasons	30.7	28.1–32.7	40.9	38.0–43.2		
7129	Carpenters and building frame and related trades workers			45.6	42.7–48.6		
7130	Electricians	16.9	15.4–18.3	31.8	29.4–34.3		
7134	Insulation workers	35.5	25.3–42.4				
7135	(FI) Glaziers			59.6	51.0–64.1		
7136	Plumbers and pipe fitters	35.4	34.2–36.3	48.0	46.1–50.0		
7150	Building structure cleaners and caretakers			40.2	37.8–42.5		
7210	Sheet-metal workers	30.7	27.1–33.4	41.0	37.0–43.7		
7212	Welders and flame cutters	30.7	28.1–32.6	40.4	37.3–42.8		
7220	Blacksmiths, toolmakers and related trades workers and operators	26.7	24.4–28.2	39.4	37.0–41.3		
7411	Butchers, fishmongers and related food preparers	44.0	39.0–48.1	68.2	66.1–69.8		
7420	(DK: NO; SE) Wood workers and related machine operators			38.6	32.5–42.7		
8120	Metal-processing plant operators	31.9	29.2–34.0	41.4	39.0–43.0		
8130	Glass, ceramics, paper and related plant operators						
8240	Wood-products machine and plants operators	30.1	26.8–32.5	43.9	42.1–45.1		
8320	Car, taxi, motorcycle and van drivers	35.2	32.8–37.1				
8323	Bus and tram drivers	43.5	42.4–44.6				
8324	Heavy truck and lorry drivers	34.5	32.9–35.8	41.6	39.1–44.4		
9110	Street vendors and related workers					42.1	31.9–47.6
9130	Domestic helpers, cleaners and related workers	29.5	28.3–30.6	40.2	37.2–42.8		
9310	Mining and construction labourers	33.2	31.4–34.9	41.3	38.8–43.9		
9320	Manufacturing and transport labourers	33.6	32.4–34.3	49.9	46.6–52.5		
9330	Transport labourers and freight handlers	32.4	31.5–33.8	48.6	48.0–49.0		

^1^ Data on cause-specific SA were not available for Denmark. FI—Finland, DK—Denmark, NO—Norway, SE—Sweden. Capital letters in parentheses before the name of an occupational group correspond to the abbreviated country name and indicate a small size of the occupational group (<500 workers) in the country.

**Table 8 ijerph-19-15674-t008:** Weighted means of country-specific excess fractions (EF, %) of the age-adjusted one-year cumulative incidence of prolonged sickness absence (SA) among women. Only occupational groups with statistically significant excess fractions in all countries are shown.

		EF of All-Cause SA		EF of SA Due to Musculoskeletal Diseases ^1^		EF of SA Due to Mental Disorders ^1^	
Code	Title	Mean (%)	95% CI	Mean (%)	95% CI	Mean (%)	95% CI
0011	Nursing and midwifery professionals and associate professionals					25.8	20.9–31.0
0021	Travel attendants and guides	25.8	21.0–28.4				
0026	Metal- and mineral-products machine operators						
0027	(NO) Chemical-products machine operators			46.8	43.1–49.3		
0028	Rubber- and plastic-products machine operators	23.0	21.4–24.6	48.1	43,5–51.4		
0029	Book- and paper- products machine operators						
0030	Agricultural or industrial machinery fitters, and mechanical and electrical equipment assemblers	23.9	21.4–24.6	43.6	42.4–44.8		
0035	Professionals and associate professionals in social work	18.2	16.3–18.7			47.9	44.8–51.2
0038	Customs, tax and related government associate professionals					25.1	23.6–26.9
0043	Machine operators of textile, fur and leather			41.2	36.1–44.0		
0044	Grocery and beverage machine operators	28.0	25.5–29.4	47.4	46.4–48.3		
2445	Journalists and other writers; radio and other announcers					35.2	33.1–36.7
2460	Religious professionals					54.9	50.0–58.3
4142	Mail carriers and sorting clerks	30.9	28.5–31.5	53.4	52.4–54.4		
5112	(FI; DK; NO) Transport conductors	44.9	38.2–48.5	51.9	45.9–55.9	48.4	38.7–53.7
5122	Cooks	25.8	24.2–25.8	42.1	40.7–43.6		
5130	Nursing and care assistants	36.5	34.1–37.4	46.6	43.1–50.3	31.8	27.3–36.6
5131	Child-care workers	29.2	26.7–30.5	31.9	28.4–35.6	30.3	26.1–34.8
5133	Home-based personal care workers	34.7	32.1–36.4	39.7	36.4–43.3		
5169	Unspecified protective service workers	26.4	20.8–29.4			28.7	21.1–33.8
5220	Shop, stall and market salespersons and demonstrators			22.3	18.3–26.6		
7411	(FI; NO) Butchers, fishmongers and related food preparers			60.3	48.7–66.4		
8120	(FI; NO) Metal-processing plant operators			54.8	50.3–57.7		
8240	(NO) Wood-products machine and plants operators	34.7	27.0–39.3	56.9	53.2–59.5		
8323	Bus and tram drivers	48.0	45.2–49.3	53.9	50.5–56.1		
9110	Street vendors and related workers					41.7	39.1–43.6
9130	Domestic helpers, cleaners and related workers	25.7	23.1–27.1	46.3	43.0–49.8		
9320	Manufacturing and transport labourers			50.9	48.7–52.6		

^1^ Data on cause-specific SA were not available for Denmark. FI—Finland, DK—Denmark, NO—Norway, SE—Sweden. Capital letters in the parentheses before the name of the occupational group correspond to the abbreviated country name and indicate a small size of the occupational group (<500 workers) in the country.

## Data Availability

The authors do not have the permission to share this third-party data. Due to data protection regulations, the data can only be accessed by individual researchers who have obtained permission to use the data through an application process. Danish data are available on the researcher access at Statistics Denmark (www.dst.dk/en/TilSalg/Forskningsservice.) The permission to use the Finnish data could be obtained from Finnish Social and Health Data Permit Authority FINDATA). (https://www.findata.fi/en/services/data-permits/). The Norwegian data, supporting reported results, are available from Statistics Norway.

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
