# Peer review of "Utilizing a Nordic Crosswalk for Occupational Coding in an Analysis on Occupation-Specific Prolonged Sickness Absence among 7 Million Employees in Denmark, Finland, Norway and Sweden"

_ijerph, 2022, doi:10.3390/ijerph192315674_

Round 1

Reviewer 1 Report

Comments to the Authors

Thank you for opportunity to review this interesting article that indentified occupations with a high incidence of prolonged sickness absence (SA) in Denmark, Finland, Norway and Sweden. The article is well-written and worthy of publication. I have only minor comments.

1.      Confidence intervals are not shown in Tables 1-4. They would make easier to compare different occupational groups. Authors mentioned in Material and methods -section that they estimated also confidence intervals for EFs.

2.      Authors could specify more closely how weighted mean of the country specific EFs have been calculated.

3.      There are two dots in some figures (concerning EFs) in Table 4.

4.      Authors have combined all three countries together in analysis concerning SA due to mental disorders and musculoskeletal diseases. Why this have been made? If there were separate analysis, possible country differences could be seen.

5.      Why authors make conclusion that prevention of cause specific work disability is especially important among those who have SA due to musculoskeletal diseases? How about those who have SA due to mental disorders? 

Author Response

  1. Confidence intervals are not shown in Tables 1-4. They would make easier to compare different occupational groups. Authors mentioned in Material and methods section that they estimated also confidence intervals for EFs.

Authors’ response: Tables 1-4 show occupational groups with rank 1-10 based on the EF of SA. These tables are primarily used to demonstrate whether the same occupational group appears among top 10 ranked occupations in several countries. The EFs values given in the tables play a secondary role to demonstrate that within the country the differences in EFs values for occupational groups with neighboring ranks often are relatively small. It is not possible to include the confidence intervals for EFs to the current format of the table. We included the confidence intervals into Tables 5 and 6 (Tables 7 and 8) in the revised manuscript).

  1. Authors could specify more closely how weighted mean of the country specific EFs have been calculated.

Authors’ response: We explained in the text how weighted mean EFs were calculated and added the formula. (Lines 291-298).

  1. There are two dots in some figures (concerning EFs) in Table 4.

Authors’ response: Numbers in Table 4 (Table 5 in the revised manuscript) were corrected.

  1. Authors have combined all three countries together in analysis concerning SA due to mental disorders and musculoskeletal diseases. Why this have been made? If there were separate analysis, possible country differences could be seen.

Authors’ response: We did not combine all three countries in the analyses concerning SA due to mental disorders and musculoskeletal diseases. The cause-specific SAs were analyzed similarly as all-cause SA. In order to reduce the number of tables, results for mental disorders and musculoskeletal diseases were shown jointly in one table. To unify the structure of the tables, we now splitted Tables 3 and 4 into two tables each. In the revised version Table 3 and Table 5 show occupational groups with rank 1-10 based on the EFs of SA due to musculoskeletal diseases among men and women, respectively. Table 4 and Table 6 show occupational groups with rank 1-10 based on the EFs of SA due to mental disorders among men and women, respectively.

  1. Why authors make conclusion that prevention of cause specific work disability is especially important among those who have SA due to musculoskeletal diseases? How about those who have SA due to mental disorders?

Authors’ response: We agree that prevention of SA due to mental disorders is also important. However, we found a larger number of occupational groups with excessive incidence of SA due to musculoskeletal diseases than due to mental disorders in all three countries. Our results suggest that SA due to musculoskeletal diseases are more attributed to the occupational exposures than SA due to mental disorders. While the SA due to mental disorders is more attributed to the workplace level, societal and/or structural factors than occupational exposures.  Due to this workplace preventive measures will have a higher potential for the prevention of work disability due to musculoskeletal diseases than due to mental disorders. We clarified the conclusions. (Lines 560-566).

Reviewer 2 Report

The aim of study was to identify occupations with a high incidence of all- cause and cause-specific prolonged SA among employees in Denmark, Finland, Norway, and Sweden and explore similarities and differences between the countries. They told to develop a Nordic crosswalk for occupational coding to be used in this and future cross-country comparative studies to increase the comparability of occupational groups across the countries. 

Major comments (only):

It is well and clearly written manuscript. However, although they have reliable and good statistical sources, the result does not produce much novelty. Still I recommend to publish the results of this study although they repeat much similar as earlier published. The variation in different countries was quite huge reflecting mostly? differences in variation of  working life in each country.

It is a pity that the coding system of occupations is not able to differentiate occupations of public sector and private sector. This division is most important in all Nordic countries, also in respect to SA. Social security and work disability practices are different in public and private sectors. Also work disability costs, expenditures of retirement and right to "fire" employees are different in the comparison of Nordic countries. Occupational Health services are different, perhaps seen as lower incidence of SA's longer than 30 days in Finland. Early intervention is recommended in Finland where larger enterprises are paying all costs of early retirements. In many other countries state will pay the responding costs etc.  The work ability support is different based on different work life practices and legislation and social security system may affect on differences of SA incidence. Also economic situation on different branches (including occupations) may affect the SA incidence.

In Introduction the authors tell "In general, the determinants of SA can be categorized into three major groups: micro (e.g., individual), meso (e.g., occupational class, industry, workplace) and macro (e.g., societal) level risk factors [1]. Previous research showed that SA is unequally distributed across different population groups with a noticeable variation by age, gender, geographical region, and occupational class [7-10]"

I would recommend to add something about differences in social security system both in Introduction and Discussion (only one sentence about this topic). Although Nordic countries have been considered similar they differ in many ways as above.

Author Response

Major comments (only):

It is well and clearly written manuscript. However, although they have reliable and good statistical sources, the result does not produce much novelty. Still I recommend to publish the results of this study although they repeat much similar as earlier published. The variation in different countries was quite huge reflecting mostly? differences in variation of working life in each country.

Authors’ response: The observed large variations in the occupational groups with excessive SA incidence in different countries could partly be due to cross-country structural and compositional differences which we were not able to control for. We made analyses for men and women separately and controlled for age differences in the composition of occupations within the countries by using age-adjusted incidence of SA. However, we were not able to control for age differences between the countries within the same occupation. We have discussed it under study limitations (Lines 542- 547).

It is a pity that the coding system of occupations is not able to differentiate occupations of public sector and private sector. This division is most important in all Nordic countries, also in respect to SA. Social security and work disability practices are different in public and private sectors. Also work disability costs, expenditures of retirement and right to "fire" employees are different in the comparison of Nordic countries. Occupational Health services are different, perhaps seen as lower incidence of SA's longer than 30 days in Finland. Early intervention is recommended in Finland where larger enterprises are paying all costs of early retirements. In many other countries state will pay the responding costs etc.  The work ability support is different based on different work life practices and legislation and social security system may affect on differences of SA incidence. Also economic situation on different branches (including occupations) may affect the SA incidence.

Authors’ response: Thank you for this comment on an issue that needs to be discussed. We discussed it under limitations. (Lines 547 -550).

In Introduction the authors tell "In general, the determinants of SA can be categorized into three major groups: micro (e.g., individual), meso (e.g., occupational class, industry, workplace) and macro (e.g., societal) level risk factors [1]. Previous research showed that SA is unequally distributed across different population groups with a noticeable variation by age, gender, geographical region, and occupational class [7-10]"

I would recommend to add something about differences in social security system both in Introduction and Discussion (only one sentence about this topic). Although Nordic countries have been considered similar they differ in many ways as above.

Authors’ response: The social security systems of countries is described in the Material and method part. We elaborated further the description and added the references on more detailed comparison of social security systems between the countries (Lines 150-188). We also added text about differences between the Nordic countries into the discussion. (Lines 457-463).

The following reference was added:

Ose, S.,O.; Kaspersen, S.,L.; Leinonen, T.; Verstappen, S.; de Rijk, A.; Spasova, S.; Hultqvist, S.; Nørup, I.; Pálsson, J.,R.; Blume, A.; Paternoga, M.; Kalseth, J. Follow-up regimes for sick-listed employees: A comparison of nine north-western European countries. Health Policy 2022; 126(7):619-31. (Ref # 29).

Reviewer 3 Report

Comments

1. The revised introduction should improve the description of the contribution of the manuscript to the literature.

2. Sick pay is a determinant of sickness absence spells (https://doi.org/10.1002/jae.2620). This issue should be stated in the paper.

3. Occupation is a choice variable for workers. Does this have implications for the interpretation of the estimates that are reported in the paper? 

4. There is substantial variation in sickness absence spells within occupational groups.

5. The composition of the workforce in terms of occupational groups is not identical in the Nordic countries.

6. Are there practical policy conclusions that stem from the results?

Author Response

  1. The revised introduction should improve the description of the contribution of the manuscript to the literature.

Authors’ response: We revised the introduction and have now further elaborated the contribution of the manuscript to the existing body of knowledge. (Lines 71-84).

  1. Sick pay is a determinant of sickness absence spells (https://doi.org/10.1002/jae.2620). This issue should be stated in the paper.

Authors’ response: We revised the manuscript according to the comment and added a few relevant references. (Lines 72-75).

The following references were added:

  • Puhani, P.,A.; Sonderhof, K. The effects of a sick pay reform on absence and on health-related outcomes. J Health Econ 2010; 29(2):285-302. (Ref # 26).
  • Böckerman, P.; Kanninen, O.; Suoniemi, I. A kink that makes you sick: The effect of sick pay on absence. L Appl Econometrics 2018; 33(4):568-79. (Ref # 27).
  • Hemmings, P.; Prinz, C. Sickness and disability systems: comparing outcomes and policies in Norway with those in Sweden, The Netherlands and Switzerland. Economics department working papers No.1601. OECD 2020. Available online: https://www.oecd.org/officialdocuments/publicdisplaydocumentpdf/?cote=ECO/WKP(2020)9&docLanguage=En (Accessed on 11 November 2022). [Ref # 28).

  1. Occupation is a choice variable for workers. Does this have implications for the interpretation of the estimates that are reported in the paper?

Authors’ response: We agree with the comment. The observed occupational differences in excess fractions of SA could be attributed not only to the differences in exposures between the occupational groups but also to the selection into the occupation. (Lines 498-500).

  1. There is substantial variation in sickness absence spells within occupational groups.

Authors’ response:  We agree with the comment. The larger the variation in SA between the employees within the occupational groups, the broader the confidence interval and less likely is the difference in SA incidence between such occupational groups and the general working population estimate statistically significant. In the cross-country comparative analyses of occupational groups with excess SA we selected occupations with SA incidence being statistically significantly higher than the respective population average.

  1. The composition of the workforce in terms of occupational groups is not identical in the Nordic countries.

Authors’ response:  This is true. To increase the comparability of occupational groups across the countries we developed a Nordic crosswalk for occupational coding. We observed the differences in the size of occupational groups across the countries with size of some occupational groups being small. To further increase the comparability of occupational groups across the countries we further merged 124 common Nordic codes with large differences in the size across the countries into 44 aggregated occupational groups. In all Tables we used the abbreviated country name to mark the occupational group with a relatively small size (< 500 workers) in that country.

  1. Are there practical policy conclusions that stem from the results?

Authors’ response: “Our results indicate that occupational preventive measures have a high potential for the prevention of cause-specific work disability, especially due to musculoskeletal diseases. The prevention of work disability could be targeted at the identified occupational groups with excess prolonged SA in all Nordic countries.”

Reviewer 4 Report

The article presents a topic of interest to the scientific community, especially through the cross-cultural approach, covering four countries.

A major problem of the study is the use of data from 2014, i.e. 8 years ago, which may make the results no longer relevant for the present.

I recommend that a solid rationale be written about the current impact of the study based on older data. At the same time, the questions should be answered: Were there no other studies done on this subject in the meantime? Is the data still current?

To improve the article, I recommend the following:

Line 16 – The abstract should be introduced by a general phrase related to the theoretical approach, the motivation of the study regarding the subject addressed.

Line 35 – I believe that the subject requires a broader approach in the theory part.

Line 61 – The research gap needs to be elaborated more thoroughly.

Line 70 – The motivation and relevance of using older data should be mentioned.

Line 381 – The research niche appears more broadly in the discussion. It can be the basis to develop the topic in the introduction.

Line 474 – To be elaborated further.

Author Response

The article presents a topic of interest to the scientific community, especially through the cross-cultural approach, covering four countries.

  • A major problem of the study is the use of data from 2014, i.e. 8 years ago, which may make the results no longer relevant for the present. I recommend that a solid rationale be written about the current impact of the study based on older data. At the same time, the questions should be answered: Were there no other studies done on this subject in the meantime? Is the data still current?

Authors’ response:  We agree that the use of data from 2015 in the analyses is an important issue that requires discussion. In all Nordic countries there are typically relatively large delays in getting the register data for research purposes, so that even newly acquired data is typically a couple of years old. In addition, getting the permission to use and combine the data from various administrative sources is a long process causing further delays in obtaining the data. Furthermore, the register data are collected for administrative purposes and cannot be used for research analyses before careful checking and cleaning. For our analyses, we have thoroughly elaborated the data by merging information from various administrative sources. When selecting the year for the analyses we also took into consideration that the national occupational classification codes are most comparable with the ISCO-88 (COM) classification. These are the reasons for why the data used for the analyses are relatively old and cannot be simply updated. The advantage is that we have harmonized data across the countries. Even though the incidence of prolonged SA may change over time, the occupational differences in the incidence are rather stable over time. In the current study, we explored the occupational difference using a relative measure (excess fraction) instead of absolute (incidence). Our main finding regarding occupations with excessive incidence of prolonged sickness absence is likely to apply more widely than to a single point of time and thus relevant also for the present time. There are no other studies done on this subject using more recent data.

We have now expanded the discussion regarding limitations of the used data (Line 551-557).

To improve the article, I recommend the following:

Line 16 – The abstract should be introduced by a general phrase related to the theoretical approach, the motivation of the study regarding the subject addressed.

Authors’ response:  Due to word limitation of the abstract (max 200) we were not able to add a general sentence related to the theoretical approach and the motivation of the study subject into the abstract.

Line 35 – I believe that the subject requires a broader approach in the theory part.

Authors’ response:  We revised the introduction according to the comment. (Lines 35-37, 44-50).

Line 61 – The research gap needs to be elaborated more thoroughly.

Authors’ response:  We have now more thoroughly elaborated the research gaps. (Lines 71-84).

Line 70 – The motivation and relevance of using older data should be mentioned.

Authors’ response:  We added the text regarding the use of older data to the Material and method part (Lines 105-107) and discussed it under limitations (Lines 551-557).

Line 381 – The research niche appears more broadly in the discussion. It can be the basis to develop the topic in the introduction.

Authors’ response:  We have revised the introduction according to the comment. (Lines 71-84, 95-99).

Line 474 – To be elaborated further.

Authors’ response:  We have further elaborated the conclusion. (Lines 560-566).

Round 2

Reviewer 3 Report

I am happy with the paper.

Reviewer 4 Report

Good luck in your future research!